# University Mentoring Programmes for Gifted High School Students: Satisfaction of Workshops

**Alba Ibáñez García** [1][iD]**, Teresa Gallego Álvarez** [1][iD]**, Mª Dolores García Román** [2]**,**
**Verónica M. Guillén Martín** [1][iD]**, Diego Tomé Merchán** [2] **and Serafina Castro Zamudio** [3],*[iD]

[1]  Department of Education, Faculty of Education, University of Cantabria, 39005 Santander, Spain;
    alba.ibanez@unican.es (A.I.G.); teresa.gallego@unican.es (T.G.Á.); veronica.guillen@unican.es (V.M.G.M.)
[2]  University of Málaga, 29010 Málaga, Spain; mdolores@uma.es (M.D.G.R.); diegotome@uma.es (D.T.M.)
[3]  Department of Personality, Evaluation and Psychological Treatment, Faculty of Psychology,
    University of Málaga, 29010 Málaga, Spain
*  Correspondence: scastro@uma.es; Tel.: +34-952132556

**Abstract:** This paper analyses the degree of participant (mentees, mentors, and technical-research team) satisfaction with two university mentoring programmes for pre-university students with high intellectual capacities in Spain. Three versions of a Likert-type scale questionnaire were applied (mentees, mentors, and technical-research team), resulting in a total sample of 43 questionnaires from mentors, 314 from mentees, and 43 from the technical-research team in 43 workshops offered by the GuíaMe-AC-UMA Programme; and 27 questionnaires from mentors, 203 from mentees, and 27 from the technical-research team in the 27 workshops offered by the Amentúrate Programme. The results indicate a high level of satisfaction with the development of the workshops offered by both programmes, on the part of all participants. No significant differences were found in terms of thematic area or gender, although there were differences in age. The participation of the three agents involved in this training offer was very successful, and our results supported the findings of previous investigations. More work is required on the transfer and maintenance of the impact that this type of programme can have on young pre-university students with high abilities.

**Keywords:** gifted secondary school students; mentoring; university context; workshop; satisfaction

## 1. Introduction

The origins of the term "mentor" date back to Ancient Greece. In *The Odyssey*, Ulysses marches off to the Trojan War, leaving his son Telemachus in the care of Mentor, who would take charge of his education and take on the role of teacher, adult friend, advisor, and protector [1]. Eventually, the term "mentor" was used to describe a person who would assume those roles and create a bond with their mentee, through which they share their experience and knowledge so that the other person can develop successfully on an academic, professional and personal level [2].

If we focus on the educational context, and, in particular, on the point of access to university, mentoring can be defined as a continuous feedback process of help and guidance between the mentor (a senior student who values the knowledge and skills required to help), and a student or a group of incoming students, in order to meet their needs and optimise their development and learning potential [3] (p. 92).

Studies indicate that all mentoring actions are based on common objectives in terms of their application: to help build or improve the set of intellectual, personal, social, and technical skills that the teacher has to convey for their students to learn [4]; to develop effective learning processes in a practical way (emphasis is placed on potential) for the acquisition of knowledge, attitudes or

skills in general, which can be transferred to socio-personal, professional and academic areas of personal development [5–9]; to improve the performance of the mentee in contexts such as academia, and improve the satisfaction of the mentors [10], including those of older age and experience in the institutions [11]; to increase satisfaction, positive attitudes and motivation regarding the work or academic environment [12]; to serve as a real alternative of learning and closer to the people involved; to serve as support in periods of transition (from one study to another, from adolescence to adulthood, incorporation into the world of work, etc.); to develop greater involvement, commitment and collaboration between the members of an institution, association or organisation, and with the institution itself; and to develop social capital, generating benefits from social connections and trust [13], facilitating new and healthier lifestyles [14].

Despite their many advantages, authors such as Klasen and Clutterbuck [15] point out that mentoring processes also have some disadvantages, especially organisational ones: the need for great support and organisational effort, limits between confidentiality and openness among the members of the process, the mentee's possible dependence on the process, difficulties in objectively measuring the results, the lack of a mentoring tradition in our country with the consequent problems of extrapolating models from other countries and cultures, and that most of the actions are carried out in informal contexts without evaluation and planning.

There are many mentoring models, including formal, informal [16], and cross-aged tutoring [17,18], that have already been applied to people with high intellectual capacities [19,20], peers, teams, liaison, e-mentoring, or a combination [21]. Formal mentoring, also known as one-time, planned, intentional, or systematic mentoring, was considered in this study. This model has been increasingly used in the last 30 years and its fundamental characteristics are as follows: structured action where objectives and expected benefits or achievements are planned; it can be carried out individually or in groups; organisers or tutors are involved as institutional agents who are in charge of selecting the mentors and supervising the process; the mentor needs training to carry out their role (sometimes the mentee also needs training); the timing is variable (there are no formulas or methods indicating the ideal period for mentoring to be successful); and, the selection of mentors is based on the specific experience they have and their personal, professional, and academic achievements.

Three agents are involved in this type of mentoring: the mentor, mentee, and tutor: (a) the mentor is a person with greater knowledge and experience who is considered suitable to help others; (b) the mentee is a person who is in a disadvantaged position who wants, on a voluntary basis, to receive help and guidance from another person with greater experience; and, (c) the tutor is the coordinator/supervisor of the process, who exercises monitoring, training, and evaluation functions. The characteristics that a mentor, mentee, or tutor should have in order to satisfactorily develop their role in mentoring programmes have been described by many authors [4,22–31]

Monitoring the process and results is indispensable for the success of any mentoring programme, as it allows the effectiveness and impact of the programme to be identified according to its objectives, as well as possible future improvements. Evaluation involves obtaining both quantitative data (through questionnaires, record sheets, psychometric tests, etc.) and qualitative data (observations, opinions, suggestions, or anecdotes). In order to carry out the planning, it is also necessary to delimit some dimensions (related to the objectives, mainly the satisfaction of the agents involved) and indicators, as well as the necessary protocols to register them, and to ensure that these cover the points of view of all the participants (mentors, mentees, and tutors). When working with children, it may also be useful to include families or counselling teams in this assessment.

Mentoring programmes in Spain, and specifically in the university environment, are divided into those created within the universities themselves and those created in non-institutional orientation programmes [32]. The main objectives of both are to facilitate the transition of new students to the university environment, although, over time, other types of assistance have been provided to students with specific needs, such as students with disabilities, those over a certain age (e.g., 25, 40, or 45), international students, practicum and final year students, postgraduate and doctoral students and,

as in the present study, students with high intellectual capacities. It should be noted that systematic studies focused on mentoring, especially those regarding high ability students, are scarce. In addition, a lack of robust methodology is often found in these studies which makes it difficult to perform a meta-analysis [1]. Currently, most Spanish universities have their own mentoring programmes, some of which are organised and coordinated in state initiatives such as the UPM Mentoring Network (2012) or Spanish Mentoring Network (REME). Few university mentoring programmes have been developed for students with high intellectual capacities [1].

Until recently, in most European countries, the main assumption was that gifted students do not have specific needs and therefore do not need any special attention. However, according to some research, the need to pay attention to the special needs of gifted students has gained more and more ground. In order to maximize the potential of gifted students and to make gifted education sustainable, it is essential to improve the flexibility of schools, diversify teaching methods and techniques, enrich curriculum content, and increase the qualifications of teachers working with gifted students [33]. Giftedness includes some aspects related to intellectual behaviours such as curiosity, concentration, persistence, problem finding, and reasoning [34]. These behaviours entail that gifted students also have special academic needs that require curricular interventions and enrichment programmes. These programmes involve modified or enriched learning experiences, which could improve students' motivation by responding to their needs [35]. In this respect, and although it may seem paradoxical, several studies have shown that the phenomenon of underachieving gifted students exists in many schools. Moreover, among the gifted population, the percentage of underachieving gifted students is quite significant. There is a need to identify effective and practical strategies that can support underachieving gifted students in school settings [36]. Yet, so far, not many studies have reported on effective practical strategies and programmes to alleviate or reverse the issue of underachievement [37,38].

Some studies also indicate that gifted students can present difficulties in the emotional area. In 2016, Perera [39] proposed the theoretical consideration that the effect of emotional management on academic performance may be attributed to cognitive, motivational, and interpersonal processes, and sometimes this emotional area has not received the attention it deserves. Learning requires that students feel comfortable in the classroom, which entails having a positive, relaxed, and stimulating environment. Emotional management should be one of the goals of education. In addition, we think that mentoring programmes for gifted students are also "very helpful in promoting a sense of relatedness and "gifted" community for gifted students' as has been evidenced by [40–42].

Brigandi and colleagues [43] found that when high ability students receive learning that matches their interests, they are more likely to perceive it as beneficial, remember the learning process in a pleasant way, and maintain an interest in the topics covered. It is thus important to identify factors that contribute to the creation of positive learning environments that support talent and mentoring programmes, among others, are appropriate for meeting the expectations of these students. An increasing number of studies also suggest that students with high abilities benefit from being considered as a homogeneous group, as is the case in this study [44–47].

Based on this, the mentoring programmes for gifted students offer an educational alternative that attempts to provide students with a cognitive stimulus as well as with a safe place to experience positive emotions and good relationships among peers and with mentors.

The GuíaMe-AC-UMA programme is used at the University of Málaga in Spain and is currently (2019) in its VIII edition. Since its beginning, GuiaME has worked from a face-to-face mentoring approach, in small groups and from rich and stimulating learning contexts. It is interesting to note that the GuíaMe-AC-UMA programme has served as a guide and orientation to mentoring for other universities, such as the University of La Laguna, the University of the Balearic Islands, and the University of Cantabria, among others.

The general objective of the present study is to identify the degree of satisfaction among mentors, mentees and the technical-research team in the first and important phase of mentoring programmes

developed at the University of Málaga and the University of Cantabria for secondary school students identified as having high intellectual capacities. As specific objectives, we propose:

(a) To check the impact of some variables (e.g., profile and number of participants, age, gender) on the degree of satisfaction obtained with the workshops.

(b) To know which aspects of the design, structure, organization, and development of the workshops are generating the highest level of satisfaction and which aspects would still need to be improved.

To this end, the following research questions are considered:

1. Are there differences in overall satisfaction with the workshops in general, and with each one in particular, according to participant profiles: mentors, mentees, and the technical-research team (TRT)?

2. What items are evaluated by the participants (mentors, mentees, and TRT) as the best and worst?

3. Are there differences in the degree of satisfaction expressed by mentees according to their gender or age?

4. Are there differences in the degree of satisfaction expressed by the mentees according to the thematic area of each workshop, or according to the number of participants?

## 2. Methodology

The GuíaME-AC-UMA and Amentúrate programmes organise their mentoring proposal in two phases. In the first phase, students/mentees are offered the opportunity to meet university teachers and researchers/mentors as a group, in a workshop format. Once a mentor/mentee match is generated, an individualised mentoring process is carried out in the second phase, which contributes to the development of a personal and/or academic project chosen by each mentee in consensus with their mentor. This individual phase was mandatory in the Amentúrate programme, and optional in the GuiaMe-AC-UMA programme, depending on whether the mentee wants to participate or not, although it is recommended to all students. During the programme's closing session, the mentees are asked to present a research paper produced with their mentor, based on their interests.

Students with high abilities or mentees who enrol in these two programmes are motivated by their interests, and the potential to study content in depth, and the challenge of training, the possibility of getting in touch with lines of research and scientific university activities, the optimal development of their professional vocation and orientation, and the improvement of their social-emotional development.

In the case of GuíaMe, in order to be able to participate in the mentoring programme, the counsellors inform the high school students with high capacities about the programme, and those interested in the programme register by sending the required documentation through the website (www.guiame-ac.es). Once they have registered, they can sign up for all the workshops they are interested in, taking into account that the criteria used to cover the places offered is the order of registration.

The Amentúrate programme was widely disclosed to the whole educative community. The students interested need to send a motivational letter as well as recommendation letters from their parents and a teacher to participate in the programme through the contact email or the website (https://amenturate.unican.es/). After the admission process, all workshops were offered to the mentees and they had to choose the workshops that they would like to attend. They also had to prioritize their choices. When the number of students was higher than the places offered in specific workshops, the priority criterion was used for the mentees selection.

Once the process of selection and acceptance of mentees in the workshops has been completed, students participate in group mentoring or the workshops that interest them the most. This allows them to see the various areas of knowledge and future professional opportunities firsthand, to discover new options or to reaffirm their initial preferences. The workshops can be an educational challenge for students since they will increase their motivation and strengthen their determination by taking them out of their comfort zones (where the challenges are too easy and they may become bored), but not

letting them reach the panic zone (where the challenges are too great and would prompt fear of failure because they exceed previous knowledge). In short, the aim of the mentor workshops is to awaken a student's sense of wonder, motivation, interest, and entrepreneurship, and allow them to search for what they want to achieve.

Mentors, through their own workshop proposals, make their respective areas of knowledge and lines of work known to mentees, facilitating learning their own interests and lines of research. The experience promotes the creation of a link between mentor and mentee according to their possible affinities. Research into the effectiveness of mentoring suggests that the following aspects stand out as key: trust, liking, supportive relationships, and complicity between mentor and mentee [48]. To this end, mentors will seek to use effective educational strategies, discovery learning, and problem-based learning; and to enable cooperative group activities and try unstructured or even open-ended activities with a variety of learning resources that also foster motivation and creativity. In short, the mentor is oriented towards research, curiosity, dialogue, experimentation, and exploration.

The technical-research team is responsible for coordinating and designing the programme, carrying it out, adapting instruments for evaluation, collecting and analysing data, and disseminating information. The TRT is also in charge of attending to the needs and difficulties that may appear on the part of the mentees, families, mentors, guidance counsellors from the institutes and other stakeholders involved in the development of the programme; extending the number of workshops to cover all areas of knowledge taught at the University of Málaga and the University of Cantabria in which the mentees show interest; and providing training in advanced skills and mentoring relationships to the mentoring faculty. In short, the TRT is in charge of managing, training, attending, supervising, and evaluating the development of the programme for its efficient functioning.

Given the importance of the first phase, the objective of this research is to analyse the group mentoring that takes place in a workshop format, considering all the variables that are involved in its development and the profiles of all participants who attend it: mentors, mentees, and TRT.

Since these programmes are promoted in the university context, the activities are designed to be carried out in the university's own facilities (i.e., classrooms, seminar rooms, computer rooms, laboratories, and offices, among others, depending on the modality of the activity) or in other institutions or places of interest (e.g., visiting parliament, a mine, an exhibition hall, monuments, a music hall, Roman ruins, etc.).

As far as possible, the mentors will be in charge of providing the equipment and/or materials needed for the development of the group and/or individual mentoring.

The present research fulfils the ethical norms of research and the legal requirements for this type of study. All participants gave their informed consent to inclusion before participating in accordance with the Declaration of Helsinki, and the protocol was approved by the Ethics Committee of the University of Málaga (CEUMA: 61-2018-H) and the University of Cantabria (CEProyecto 4/19) which guarantees the confidentiality and anonymity of the data, among other fundamental ethical aspects.

*2.1. Participants*

The study included three groups of participants who, on a voluntary basis, participated in one of these mentoring programmes based on its social and personal benefits: mentees, mentors, and the technical-research team. All our data come from an incidental sample and are limited to the participants of both programmes developed during a course.

2.1.1. Mentees

Mentees are gifted students in high school. The selection made of secondary education students was due to the fact that these are years in which students consider their vocations, as well as stages in which the motivation for study may diminish and greater school failure may appear [49].

A total of 130 students with high intellectual capacity participated in the GuíaMe-AC-UMA programme, from 3rd year ESO (Obligatory Secondary Education) to the last year of high school.

Regarding the gender, 61.54% (n = 80) were male and 38.46% (n = 50) were female. At the beginning of the programme, all the mentees were between 13 and 18 years old. The percentage of mentees per course were as follows: 38.46% were in third ESO, 33.08% in fourth ESO, and 15.38% and 13.08% in the first and second course of high school respectively, before University.

With respect to the Amentúrate programme, a total of 41 students with high intellectual capacities participated, ranging from Obligatory Secondary Education and students in the penultimate year of high school. Thirty-nine percent (n = 16) were female and 60.9% (n = 25) were male. At the beginning of the programme, all the students were between 11 and 16 years old, distributed in different courses: 31.7% were in first ESO, 21.9% in second ESO, 19.5% in third ESO, 9.7% in fourth ESO, and 17% in penultimate year before University.

In both programmes, more than 80% of the students attended more than one workshop, and the average attendance at each workshop was 7–8 participants (in GuíaMe-AC-UMA, $M = 7.32$ and in Amentúrate, $M = 7.57$). All mentees attending workshops must complete the satisfaction questionnaire. In total, 314 questionnaires were administered in Málaga and 203 in Cantabria.

### 2.1.2. Mentors

The mentors were university professors and/or researchers in both programmes who participated voluntarily and neutrally, motivated by the development of talent and by developing attractive thematic workshops related to their area of knowledge and/or line of research.

Thirty-nine mentors participated from GuíaME-AC-UMA, of whom 48.72% were women and 51.28% men, and they conducted a total of 43 workshops (some mentors proposed more than one workshop). A total of 43 questionnaires were collected, one per workshop.

Twenty-five mentors participated from Amentúrate, 36% of whom were women and 64% men, conducting a total of 27 workshops. A total of 27 questionnaires were collected, one per workshop. Where more than one mentor led an activity, a single questionnaire was completed jointly.

### 2.1.3. Technical-Research Team (TRT)

The team from the GuíaMe-AC-UMA programme was made up of professors from the Faculty of Psychology and the Faculty of Science, and professionals in psychology and education. The team from the Amentúrate programme was made up of professors from the Department of Education.

There were nine participants in the TRT from GuíaMe-AC-UMA, including collaborators who were graduates in psychology or pedagogy (four members of the TRT and five collaborators). The team was made up of 77.78% women and 22.22% men.

Ten professors and one research technician, all from the Department of Education of the University of Cantabria, participated in the TRT of the Amentúrate programme. The team was composed of 81.82% women and 18.18% men.

In total, 43 questionnaires were answered by the TRT for Málaga, and 27 for Cantabria.

### 2.2. Instruments

The semi-structured assessment questionnaire prepared ad hoc by García-Román and colleagues [50] was used for the evaluation of workshops in both programmes, in its three versions: the semi-structured assessment questionnaire for the mentee; assessment questionnaire for the mentor, and assessment questionnaire for the technical-research team.

The different versions maintain the same structure, which quantitatively evaluates a total of 30 items on a Likert scale from 1 to 4 (1 = No, never; 2 = Almost nothing; 3 = Yes, quite a lot; 4 = Yes, a lot; NA = Not applicable). The questionnaire thus evaluates the level of satisfaction with the workshops as regards: usefulness, methodology, organisation and resources, teaching characteristics, and interests and expectations; as well as questions related to the subject, such as the interest generated, the relevance of the topic, and the depth or rigour.

Information was collected at the end of each workshop in which the three agents involved: the mentees, mentor, and TRT completed the questionnaires, which were collected by those responsible for each of the programmes; that is, by the member of the technical-research team or the collaborator in charge of supervising the workshop.

### *2.3. Statistical Analysis*

The data obtained from participants was analysed using the statistical program SPSS ver. 22 (IBM, Armonk, NY, USA) [51]. The confidence level was 99%.

A descriptive analysis was conducted, and the results indicated that the samples from both programmes followed an asymmetric negative distribution (i.e., high scores predominated). The Kolmogorov-Smirnov goodness-of-fit test performed to test the normality of the data evidenced a violation of the normality assumption in all cases ($p < 0.001$), so it was necessary to use a non-parametric test in the inferential analysis. The Kruskal–Wallis H test was used to compare three or more independent samples (i.e., differences between the mentors, mentees, and TRT and between mentees of different ages); the Mann–Whitney U test was used to compare two groups (e.g., satisfaction differences in the two programmes and between males/females in the case of mentees), and finally a post hoc test when the H was significative statistically.

The areas of knowledge with the best and worst evaluations were also identified and a correlation analysis (r) was performed between each item and general satisfaction using Spearman's rho.

## 3. Results

### *3.1. Satisfaction with Group Mentoring/Workshops in General and in Particular*

The average level of satisfaction with all the workshops was analysed according to the profile of the participants. To do this, an analysis was carried out differentiating between general satisfaction and satisfaction by items, in order to determine which issues were the best and worst rated. A Likert scale was used (1 = No, never to 4 = Yes, very much), that is to say from low satisfaction to a high degree of satisfaction.

The agents all evaluated the workshops with very high scores in both the GuíaMe-AC-UMA programme (M = 3.73; dt = 0.50) and the Amentúrate programme (M = 3.74; dt = 0.46), with no significant differences between the two programmes (U = 50447; $p = 0.95$). The scores for general satisfaction were very high in all three groups (with average scores above 3.5 out of 4). It is worth noting that the highest scores for satisfaction were given by the TRT in both cases, followed by the mentees and then the mentors, however, no significant differences were found in the overall satisfaction between mentors, mentees and TRT in either programme ($p > 0.01$ in both cases). These results are shown in Figure 1.

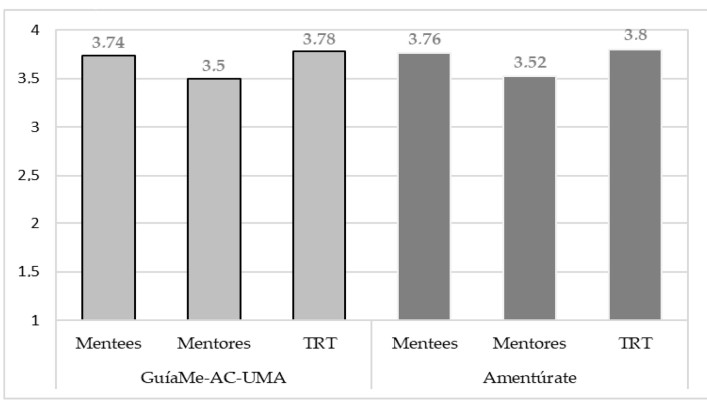

**Figure 1.** Overall satisfaction mean scores for each group.

Table 1 identifies and sets out the items that generated the most and least satisfaction, noting that the items that caused the least satisfaction are still fairly well rated, with an average score above 3, except for Items 4 (mentees) and 12 (mentors and TRT) in the GuíaMe-AC-UMA programme and Item 9 (mentors) in the Amentúrate programme, which had scores below 3.

**Table 1.** Items rated by workshop participants (group mentoring).

| | | Top Rated Items | Least Valued Items |
|---|---|---|---|
| **GuíaMe-AC-UMA** | *Mentees* | 21—The mentor tried to maintain order/discipline within a pleasant and respectful environment ($M$ = 3.92) <br> 25—I would propose this workshop for the next series of workshops ($M$ = 3.78) <br> 18—The mentor was able to resolve the doubts we raised and has been accessible ($M$ = 3.75) | 4—I believe the workshop has served to clarify my future studies at the university ($M$ = 2.98) <br> 1—I will be able to apply the knowledge acquired in secondary school to my life … ($M$ = 3.27) <br> 9—The mentor used scientific method throughout the workshop ($M$ = 3.35) |
| | *Mentors* | 25—I would propose this workshop for the next series of mentor workshops ($M$ = 3.80) <br> 24—I think that the TRT coordinating the programme gave adequate support for the optimal development of the activity ($M$ = 3.76) <br> 19—I have promoted the participation of students during the development of the workshop ($M$ = 3.69) | 12—We gave students the appropriate documentation and/or materials for the development of the workshop ($M$ = 2.73) <br> 9—I used the scientific method throughout the workshop ($M$ = 3.24) <br> 1—The mentee will be able to apply the knowledge acquired in the secondary school in their life ($M$ = 3.24) |
| | *TRT* | 18—The mentor was able to resolve the doubts we raised and has been accessible ($M$ = 3.93) <br> 20—The mentor has done their best to use simple language to explain the ideas and concepts they wanted to convey ($M$ = 3.93) <br> 21—The mentor tried to maintain order and discipline within a pleasant and respectful environment ($M$ = 3.91) | 12—The students were given the appropriate documentation and/or materials for the development of the workshop ($M$ = 2.49) <br> 1—The mentee will be able to apply the knowledge acquired in the IES, in their life … ($M$ = 3.12) <br> 4—I believe that the workshop has served to clarify the mentee's future studies at university ($M$ = 3.16) |
| **Amentúrate** | *Mentees* | 15—The number of participating students has been appropriate for the development of the workshop ($M$ = 3.81) <br> 21—The mentor tried to maintain order and discipline within a pleasant and respectful environment ($M$ = 3.80) <br> 25—I would propose this workshop for the next series of workshops ($M$ = 3.78) | 4—I believe that the workshop has served to clarify my future studies at the university ($M$ = 3.11) <br> 9—The mentor used scientific method throughout the workshop ($M$ = 3.31) <br> 5—The workshop has awakened my interest in continuing to deepen and investigate the subject ($M$ = 3.39) |
| | *Mentors* | 19—I promoted the participation of students during the development of the workshop ($M$ = 3.89) <br> 24—I think that the TRT coordinating the programme gave adequate support for the optimal development of the activity ($M$ = 3.75) <br> 18—The mentor was able to resolve the doubts we raised and has been accessible ($M$ = 3.75) | 9—I used the scientific method throughout the workshop ($M$ = 2.93) <br> 3—I made a summary or undertook reflection at the end of the workshop to strengthen the ideas ($M$ = 3.04) <br> 8—The methodology used in this workshop has been fundamentally practical ($M$ = 3.18) |
| | *TRT* | 24—I think that the TRT coordinating the programme gave adequate support for the optimal development of the activity ($M$ = 4) <br> 18—The mentor was able to resolve the doubts we raised and has been accessible ($M$ = 3.96) <br> 21—The mentor has tried to maintain order and discipline within a pleasant and respectful environment ($M$ = 3.96) <br> 25—I would propose this workshop for the next series of workshops ($M$ = 3.96) | 1—The mentee will be able to apply the knowledge acquired in secondary school in their life … ($M$ = 3.22) <br> 8—The methodology used in this workshop was deeply practical ($M$ = 3.48) |

It is worth mentioning that, in general, most of the items with the best ratings involved the interests and expectations of the mentees and the characteristics of the mentors. The lower ratings referred more to the usefulness of the workshop outside the programme and the methodology used. As can be seen in Table 1, many of the items extracted coincide in their high and low ratings among both programmes and among informants.

In both programmes, mentee and TRT scores were higher than mentor scores in the items of the questionnaire. Mentee scores tended to be higher than TRT scores in items related to the utility of the workshops (items from 1 to 5). In contrast, TRT scores were higher in items related to methodology (from 6 to 10) and teaching characteristics (from 16 to 21). In items 11 to 15, which refer to organization, TRT scores were higher than mentees and mentor scores in Amentúrate. It must be noted however that in GuíaMe-AC-UMA, mentees gave the highest scores in this part. In the last items (from 22 to 25), which are related to interests and expectations, in GuíaMe-AC-UMA best scores were given by mentees whereas in Amentúrate higher scores were from TRT. Significant differences were found in some items (Table 2).

**Table 2.** Significant results of the comparative analyses between participants (Kruskal–Wallis).

|  | Items | $X^2$ | $p$ | Significant Differences Post Hoc ($U$) |
|---|---|---|---|---|
| **GuíaMe-AC-UMA** | 12—Mentor gave the appropriate documentation and/or materials for the development of the workshop | 55.058 | 0.000 | Mentee-Mentor ($p = 0.000$) TRT-Mentor ($p = 0.000$) |
| | 16—Mentor has shown contents using attractive explanations | 11.053 | 0.004 | Mentee-Mentor ($p = 0.001$) TRT-Mentor ($p = 0.001$) |
| | 20—The mentor has done their best to use simple language to explain the ideas and concepts they wanted to convey | 10.496 | 0.005 | Mentee-Mentor ($p = 0.001$) |
| | 21—The mentor tried to maintain order and discipline within a pleasant and respectful environment | 11.731 | 0.003 | Mentee-Mentor ($p = 0.001$) TRT-Mentor ($p = 0.002$) |
| **Amentúrate** | 4—I believe that the workshop has served to clarify mentees future studies at the university | 10.610 | 0.005 | Mentee-Mentor ($p = 0.008$) |
| | 11—Organization and tools using in this workshop are appropriated | 10.003 | 0.007 | TRT-Mentor ($p = 0.003$) |
| | 12—Mentor gave the appropriate documentation and/or materials for the development of the workshop | 13.114 | 0.001 | Mentee-Mentor ($p = 0.000$) TRT-Mentor ($p = 0.000$) |

Finally, in order to understand which specific characteristics of the workshops were more related to the degree of satisfaction of the mentees, a criteria analysis was carried out to determine which specific items of the questionnaire correlated most closely with the score for general satisfaction with the workshop that the mentees marked. These results show that the mentees fundamentally value the teaching characteristics and the methodology used by the mentor. For GuíaMe-AC-UMA, the item that correlates most highly with satisfaction in the workshops was Item 17, "The mentor has managed to maintain interest and adapt the session based on our requests" ($r = 0.500$; $p < 0.01$); on the other hand, the items that correlate most highly with the general satisfaction of the mentee are: Item 19, "The mentor encouraged our participation during the development of the workshop" ($r = 0.463$; $p < 0.01$) and Item 10, "The mentor stimulated our curiosity through unstructured, discovery, or demonstration activities" ($r = 0.405$; $p < 0.01$).

*3.2. Valuation of Mentees by Areas of Knowledge*

The mentees are the true protagonists of this experience and they therefore deserve a special analysis of additional variables regarding which aspects they valued more highly, and if there is any variable that has a differential impact. The best evaluations of the workshops in the GuíaMe-AC-UMA programme by thematic area were for Education, Nursing, Music, and Public Speaking (M = 4), and the best ratings in Amentúrate were for Mathematics and Psychology (M = 4). It is not possible to indicate those valued least since the minimum score per area was above 3.

*3.3. Valuation of Mentees According to Age and Gender*

We performed means contrast tests to determine whether there were differences in satisfaction depending on gender and age variables.

No significant differences were found according to gender for any of the items in either programme, and $p > 0.01$ was found in all the contrasts executed.

No significant differences were found according to age for any item in GuíaMe-AC-UMA (M = 15.06 years, dt = 1.20). For Amentúrate, which had younger participants in the workshops (M = 13.04; dt = 1.41), the contrast test carried out found a $p < 0.01$ for Item 1 "I will be able to apply the knowledge acquired in Secondary School, in my life", it was possible to verify using a post-hoc test that the average score for this item was only significantly lower in the group of children who began the programme at 11 years of age (1st ESO). It should be borne in mind that, pupils participate from 3rd ESO to the last level before University in the GuíaMe-AC-UMA programme, but participate from 1st ESO in the case of Amentúrate.

## 4. Discussion

An important limitation to this type of educational experience aimed at students with high abilities, noted at the start of this study, is that they usually lack a means of formal evaluation to assess their effects [52,53]. The purpose of our research has therefore been to assess the degree of satisfaction generated by the mentor workshops carried out in two university mentoring programmes aimed at pre-university students with high intellectual capacities. These programmes were widely disseminated both at the university and in the community. Mentors, mentees, and TRT participated voluntarily in the programme. All our data come from an incidental sample and the results of the study cannot be directly extrapolated to other populations.

The first interesting result is that the average score for satisfaction with regard to group mentoring was very high, in both the GuíaMe-AC-UMA programme and in Amentúrate, which can be seen as positive and useful with regard to the proposal for integral training related to students with high capacities. The workshops were very well received, both those aimed at providing students with new technical and scientific knowledge, and those linked to the social-emotional or creative development of minds, results that are related to the meta-analysis of mentoring programmes conducted by DuBois and colleagues [54], noting that there is a small but significant positive effect for students regardless of whether these programmes are linked to more general objectives, such as aspects of psychosocial development or specific objectives of instrumental learning. Furthermore, they suggest that the effects of the programmes increase significantly when the programmes have certain characteristics, such as the supervision of programme implementation, selection of potential mentors, matching under specific criteria, training, the existence of a support group for mentors, structured activities, parental support or involvement, and expectations of both frequency of contact and duration of relationships, promoting the establishment of strong relationships. We found the same results on effectiveness in another meta-analysis of youth mentoring research conducted later by DuBois and colleagues [55] and published in 2011, again highlighting the effectiveness of mentoring in social, emotional, behavioural, and academic areas. They also refer to the effectiveness of these programmes, although they point out the difficulty of analysing the results and undertaking meta-analyses due to the problem of poor

methodological organisation, or the various existing concepts of what mentoring is. It should be noted that although we did not directly evaluate the effectiveness of the programme in our study, we can conclude that the high degree of satisfaction presented by mentors, mentees, and TRT can be considered an indicator of the effectiveness of both programmes.

Once it was possible to see that satisfaction with the programmes was high, it was necessary to carry out a detailed analysis by item so as to assess the elements on which this satisfaction is based.

One of the aspects most highly valued in both programmes, on the part of the TRT and the mentees, has to do with the skills of the mentor, highlighting their ability to manage the classroom and establish a good environment. ((21) "The mentor has tried to maintain order/discipline within a pleasant and respectful environment.") In this line, several studies indicate that one of the most important variables in the effectiveness, and their satisfaction with mentoring programmes is the mentor's skills and ability to foster a good classroom environment [48], or the organisational aspects connected with the timeliness of attendance at meetings, generating supportive, trusting, and stimulating relationships among participants, or having clear objectives [56]. Langhout and colleagues [57] found that the results for mentoring were more favourable when young people not only felt supported by the mentor, but also when there was a degree of structure in their relationships, aspects present in both programmes. To this end, a workshop format that, on the one hand, allows the structure of some sessions and, on the other hand, facilitates the relationships between mentors and mentees in a relaxed atmosphere that enables the learning and well-being of the participants, is proposed.

The importance of the relationship with the mentor and the mentor's accessibility is key in studies on mentoring [58], and our results for both programmes and from different informants, show the value of this ((18) "The mentor has been able to allay the doubts we have raised and has been accessible") and ((19) "I have favoured the participation of the students during the development of the workshop"). In the GuíaMe-AC-UMA programme, the TRT also highlights the role of the mentor at a communicative level ((20) "The mentor has done their best to use simple language to understand the ideas and concepts that they wanted to transmit"). Ramírez [56] emphasises that one of the functions of a mentor must be to inspire, encourage, and empower their students, to serve as a springboard for them to explore the world and to guide them in the search for answers and solutions, generating a necessary stimulus to foster the development of the mentee [25].

Another aspect very much appreciated by the different participants in the two programmes (mentors and mentees in GuíaMe-AC-UMA, and mentees and TRT in Amentúrate) involves satisfaction with the programme in general and their wish that it be carried out again in the future ((25) "I would propose this workshop for the next series of workshops"). Not surprisingly, several participants reflected their interest in recommending the programme, and not only the mentees, but also the other participants, since numerous studies indicate the benefits of mentoring in learning, facilitating the development of a personal relationship, the personal gratification of participating and the improvement of management skills that mentoring entails [59]. The literature also suggests benefits at the personal level, such as increased self-reflection since, as noted Lopez-Real and Kwan [60], mentors perceive themselves as role models and feel compelled to re-evaluate their own teaching approaches, techniques, attitudes, and so on, more deeply and critically than they would normally do. They also learn from innovative ideas and strategies that enable them to renew their teaching work [61]. Perhaps the mutual benefit from mentoring, which was also found by McDaniel and Besnoy [20], is reflected in some aspects that were assessed in our study, such as the high level of satisfaction with the programme on the part of all the agents involved, mentees, mentors, and TRT, and their support and involvement in ensuring that the programme continues.

Among the characteristics most valued by the mentors in both programmes and by the TRT of GuíaMe-AC-UMA was the support of the Technical Team during the activities ((24) "I think that the TRT which coordinates the programme has given adequate support for the optimal development of the activity"). Iucu and Stingu [62] emphasise the importance of training and support for the mentor, suggesting that both aspects are important for the development of the mentoring. Similarly,

we highlight again the results found by DuBois and colleagues [54], who point out that the importance of providing some kind of guidance for mentors and having continuous support at their disposal are key aspects in the effectiveness of mentoring.

It should be noted that the average number of 7–8 participants per workshop was valued very positively at the Amentúrate ((15) "The number of participating students has been appropriate for the development of the workshop"). This may be because this number of participants allows for interaction between them and also facilitates a stronger relationship with the mentor when it comes to asking questions or in practical activities. Carrying out activities in groups is of particular importance for high ability students, as several studies emphasise their need to be able to share experiences and interests with similar peers so that they do not mask their high abilities for fear of social rejection [46,63–65] and to promote social skills [66–68].

With regard to the absence of significant differences in satisfaction according to the age of the participants, the design of the program has respected the differences in capacity, interests, motivations and competencies of the students, becoming a real experience of talent development through a very personalized learning [69]. These results encourage us to continue working and addressing this issue in depth in order to investigate some of the data obtained.

A variety of themes were chosen for the workshops. Several factors that may influence their assessment, however, such as the relationship between the mentor and mentees, the climate generated among the mentees, or their own initial interest in the subject. Furthermore, both universities the workshops aim to satisfy the needs of the mentees at a cognitive level. The workshops try to take students out of their comfort zone, allowing them to fully develop their potential, performing tasks that pose a challenge, and creating an environment that supports their efforts and acknowledges their successes; important aspects to be taken into account, as indicated by different studies [43,45,70].

On the other hand, both programmes coincide in many of the items that can be improved by almost all the evaluation agents. These items involve aspects related to the applicability of knowledge for the future ((4) "I believe that the workshop has served to clarify my future studies at university"), ((1) "The mentee will be able to apply the knowledge acquired at the secondary school in their life"). This may be due to the age of the mentees, since it ranged from 11 years old and the knowledge they acquire during the workshops may not correspond to the reality of their educational centres or their future ideas about which studies to choose. In fact, significant differences by age are only found in the Cantabria programme among the youngest mentees, aged 11. Some studies have highlighted the positive effects of mentoring on the academic and vocational results of the mentees [71], pointing out that being able to participate in real work contexts has the potential to affect vocation, interests, and educational commitment by offering opportunities to explore interests and feel competent in a real environment [72]. Younger participants may not perceive the benefit of the activities for their vocational development because that is still in the distant future.

In addition, the study also identified the need to continue addressing the methodology used throughout the workshop. The least valued items involved a lack of perception of the use of the scientific method during the workshops ((9) "The mentor used the scientific method throughout the workshop"), an aspect that coincided in the mentors and the mentees of both programmes, which may be related to the fact that several of the workshops offered were in-depth and involved dynamic, playful, or masterful activities, but did not require the scientific method to achieve their objective. Some of the workshops were linked to other areas, such as art, music, oratory, and study techniques, which do not lead to scientific activities as such. It may also be related to the methodology used in the workshops ((12) "The mentor has given the student the appropriate documentation and/or materials for the development of the workshop"), about which the mentors and TRTs agreed in the GuíaMe-AC-UMA programme, but not the mentees, which suggests that perhaps the latter were more satisfied with the practice of the workshop without the need for additional materials or documentation, an aspect that perhaps concerns the mentors more.

In any case, the mentoring experiences developed at both the University of Málaga and the University of Cantabria were very satisfactorily valued. The more extensive experience of the University of Málaga in this first phase of the programme is corroborated by the results for the University of Cantabria, which is in its second series. In spite of some issues that need further improvement, it seems that the manner in which is the workshops were carried out is interesting, and beneficial for all the participants for different reasons, as it is also found in similar studies [20].

## 5. Conclusions

The programmes GuíaMe-AC-UMA and Amentúrate can be considered positive and useful to achieve integral formation for students with high capacities.

The high satisfaction found in the workshops of both mentoring programs is linked to aspects such as:

(a)  Mentor's skills, mentor ability to foster a good classroom environment, and mentor's accessibility, a factor that enables the learning and well-being of the participants.
(b)  The importance of training and having continuous support for the mentor is a key for the effectiveness of the mentoring.
(c)  Group mentoring developed in small groups (7–8 participants), promoting social interaction, and facilitating a stronger and deeper relationship between mentor and mentees.
(d)  The mutual benefit from mentoring. Mentors and mentees value positively the continuity of the program, so it appears that they themselves recognize the personal benefits obtained from the workshops and desire them for others.

Both programmes should continue to work on improving the aspect related to the applicability of knowledge for the future, encouraging the linking of the experience in these programmes with the vocational development despite their youth. We think that a second phase based on individual mentoring could help to achieve this question.

Finally, we are convinced that experiences like GuiaMe-AC-UMA and Amentúrate guarantee the fulfilment of the University Social Responsibility and help to make visible the possibility of developing talent from a process focused on the differences in capacity, interests, motivations, and competencies of the students, becoming a real experience of talent development.

**Author Contributions:** Conceptualization, A.I.G., M.D.G.R., D.T.M., and S.C.Z.; data curation, T.G.Á., V.M.G.M., and S.C.Z.; formal analysis, V.M.G.M.; funding acquisition, A.I.G., V.M.G.M.; investigation, A.I.G., T.G.Á., M.D.G.R., V.M.G.M., D.T.M., and S.C.Z.; methodology, A.I.G., V.M.G.M., and S.C.Z.; project administration, A.I.G. and S.C.Z.; resources, A.I.G., T.G.Á., M.D.G.R., D.T.M., and S.C.Z.; supervision, A.I.G. and S.C.Z.; validation, A.I.G. and S.C.Z.; visualization, A.I.G. and S.C.Z.; writing—original draft, A.I.G., T.G.Á., M.D.G.R., D.T.M., and S.C.Z.; writing—review and editing, A.I.G., T.G.Á., M.D.G.R., D.T.M., and S.C.Z. All authors have read and agreed to the published version of the manuscript.

**Funding:** GuiaMe-AC-UMA received no external funding. The Amentúrate programme is funded by a university grant for subsidized by SODERCAN/FEDER (11.JE04.64661).

**Acknowledgments:** Thanks to the mentors, the students and their families, as well as the University of Málaga, the Delegation of Education of Málaga, and the University of Cantabria for supporting this project. The authors would like to acknowledge Enrique Viguera, principal investigator of the scientific dissemination program "Encuentros con la Ciencia", financed by the Spanish Foundation for Science and Technology, and its support to the GuíaMe-AC-UMA programme. We would like to thank the other colleagues of Amenturate (UC), especially Juan Amodia and Alicia Gallardo, and the Cantabrian Association for the Support of High Abilities (ACAACI) for their contribution in our investigation of the Mentoring Programme: Amentúrate.

**Conflicts of Interest:** The authors declare no conflict of interest.

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
