# Peer review of "University Mentoring Programmes for Gifted High School Students: Satisfaction of Workshops"

_sustainability, doi:10.3390/su12135282_

Round 1

Reviewer 1 Report

The present work deals with a subject of great interest as it is the one of the students with high capacities. The methodology used is correct and, overall, the research design is acceptable. However, in our opinion, it presents different deficiencies that must be corrected in order to consider the possibility of publishing this manuscript. We refer below to the main ones:

  1. Although the introduction correctly establishes the justification for the work, and why this research has been undertaken, only general goals are presented as research questions and the presentation of a very general objective (page 3). However, we miss a later section in which the objectives and sub-objectives of the work are detailed and precise. That is, in a concrete way, what is today a standard in any scientific research based on field work, as is the case here.
  2. The sample is presented in the methodology section, offered as a description of the participants (p. 5). However, we would need to go deeper into the representativeness of the sample. Even though this is a case study, it is always advisable to establish to what extent the results can be applied in other similar contexts.
  3. The presentation of the results seems to us to be poor. We consider that it should be more extensive and detailed, especially in the graphic section. We believe that this section needs to be considerably improved.
  4. Although there is a wide discussion, and a glimpse of conclusions implicit in them, this manuscript lacks a specific section devoted precisely to conclusions. It is essential that the authors rigorously determine what conclusions they have reached, and these should be set out in a specific and independent section.
  5. Another of the aspects of this work that can clearly be improved are the sources used. As can be seen from the bibliographical references, there is only one source corresponding to 2018 and another to 2019, from a total of 60 works consulted. There is a very high percentage that are a decade or more old. In a subject where so many valuable contributions are being made in recent years, it is not acceptable to work with such old sources. Precisely when we read that it was a research on university tutoring for pre-university students with high intellectual capacities, we thought that it could be the application in Europe of many researches that have been made recently in the United States. However, as we say, only two works are collected after 2018. The consultation of sources on this important issue must be thoroughly reviewed.
  6. There are also some defects in the format and layout of the article (in the tables -as in table 1-, in the line spacing -very remarkable on p.3, for example-, etc.). Although in most cases they are minor defects in table 2, however, an alternative design should be sought. The text that appears vertically is difficult to read, and makes it difficult to consult this table. On p. 13 there is also a significant mismatch where the contribution of the co-authors appears, where both initials and full names are given.

Author Response

Please reconsider the enclosed manuscript, titled “University Mentoring Programmes for Gifted High School Students: Satisfaction of Workshops” (previously titled as ‘University Mentoring Programmes for Gifted High School Students: Impact of Workshops’) by Alba Ibáñez García, Teresa Gallego Álvarez, Mª Dolores García Román, Verónica M. Guillén Martín, Diego Tomé Merchán & Serafina Castro Zamudio for publication as an original article in Sustainability.

All suggestion proposed by reviewers have been accepted. No contents have been removed from original article, but we have added several sentences with more explanations as well as tables and graphics to make the article easier to understand. All additions are highlighted in red along the document. Minor errors have also been detected and their corrections are also marked in red. We attach specific documents with the responses for each reviewer. reviewer.

We believe that this manuscript has highly improved thanks to reviewers’ suggestions and is now more appropriate to be published by the journal Sustainability. We remind you it is a novel study that takes the perspective of the efficacy of mentoring programmes and the implications of our study can be considered to improve professional practices with students with high capacities.

As we already confirm in the previous submission, this study has not been published elsewhere or it has not been submitted simultaneously for publication elsewhere. This paper has been professionally proofread. All the authors agreed to send the article with these modifications.

We are sincerely grateful for your time given to re-examine this study.

Sincerely yours,

Serafina Castro

Reviewer 2 Report

The work is well organized but the bibliographical part should be better supported. In the sense that more recent bibliographic sources (range of years 2017-2019) that support such an innovative research should be added. The results should be further expanded to support the salient features of the study.

Author Response

(The authors gave the same response as above.)

Reviewer 3 Report

This was an interesting and well-written article. I appreciate the authors’ efforts to design and administer this program, and I am encouraged to see this type of ongoing program evaluation in practice. After reading the manuscript, I have a few questions/suggestions that may help strengthen the paper further.

First, while the authors note this is a descriptive paper and refrain from using causal language (beyond the use of “Impact” in the title, which potentially would be more accurate as “Satisfaction with workshops”), I think there should be more of a discussion of the selection issues at play in the study. First, students are filtered in some way based on their achievement to be invited to participate in the mentoring program. Then, students choose whether or not to participate. Finally, students choose whether and which workshops to attend. I think it’s important the authors acknowledge these multiple layers of selection and how that could be affecting student satisfaction with the mentoring.

On a related point, I think it would be helpful to have more detail about how many workshops were offered and how many, on average, students actually attended. Including take-up rates like this would help show how selected the students responding to the surveys about their satisfaction with the workshops actually are.

Second, I’d like more clarification about the conceptual framework and the targeting of the program. Students in the program are ages 11-16, which is a pretty big range. What was the rationale for working with students at these ages? For an international audience, how do these ages/grades fit into the sequence leading to college? What, if any, sort of differentiation of was there in terms of content and facilitation of workshops for students of different ages, and did the mentors get to choose which ages they worked with? Similarly, why the focus on high-achieving students? Any literature discussing the impacts of mentoring for students with different achievement backgrounds would be helpful to include.

Third, with the methods, I’m curious about the choice to look at items individually, rather than looking for latent constructs that could aggregate items (for example, there seem to be items related to satisfaction with the content of workshops, delivery of workshops, and mentors specifically). Doing this type of analysis could make patterns in satisfaction more readily apparent.

Finally, two small points. First, the formatting for Table 2 is a little distracting; I think it would be helpful if the first two columns had a larger width to fit the text. Second, line 203 starts with “With respect to” and I think it’s missing something- perhaps “With respect to student characteristics”?

Overall, I think this is a strong paper, and I look forward to reading the final version.

Author Response

(The authors gave the same response as above.)

Round 2

Reviewer 1 Report

Dear Editor and Authors,

The article has been substantially improved, congratulations on the effort made.

There are still some minor considerations to be made, but what concerns me most are the bibliographical references, which need to be reviewed in greater depth.

I give only two examples, where data appear in Spanish, as in reference 50 which says "Documentos no publicadas" or with an incorrect format, as in 51. These are just two examples, so please check all the references.

It would also have been appropriate to add some even more recent references. Although the authors have done a good job of updating, references older than 5 years are still the majority.

Author Response

RESPONSE TO REVIEWER 1 COMMENTS (ROUND 2)
The article has been substantially improved, congratulations on the effort made.
1. There are still some minor considerations to be made, but what concerns me most are
the bibliographical references, which need to be reviewed in greater depth.
I give only two examples, where data appear in Spanish, as in reference 50 which says
"Documentos no publicadas" or with an incorrect format, as in 51. These are just two
examples, so please check all the references.
Response 1: References 21, 50 and 51 have been amended. We have carefully reviewed
all the references. The result is:
21. Harrington, A. E-Mentoring: The Advantages and Disadvantages of Using Email to
Support Distance Mentoring. European Social Foundation. Available online:
http://www.coachingnetwork.org.uk/informationportal/
articles/ViewArticle.asp?artId=63 (accessed on 1 April 2020).
50. García-Román, M.D.; Castro-Zamudio, S.; Tomé Merchán, D. Escala para la
valoración de las actividades de talleres mentorinteruniversitarios: versión
estudiante, versión mentor y versión ET; Unpublished documents; Universidad de
Málaga: Málaga, Spain, 2018.
51. SPSS. Statistics for Windows, Version 22.0; IBM Corporation: Armonk, NY, USA,
2013.
2. It would also have been appropriate to add some even more recent references. Although
the authors have done a good job of updating, references older than 5 years are still the
majority.
Response 2: Based on your first comments, we did a search on WOS and SCOPUS and
we incorporate new articles in both English and Spanish. we also found some articles
in Korean that we discarded because of the language.

Reviewer 3 Report

I appreciate the authors' responsiveness to the reviews; I think this is an improved version of the paper. However, I have two concerns that I think still need to be addressed. 

First, and most importantly, I would still like a little more information about the number of workshops students attended- there are 171 students and 157 questionnaires, which is an average of about 3 sessions per student. The authors also note that 80% of students attended more than 1 session. Is the distribution pretty tight around 3 sessions? Or are there students who extend out to the tail (there were over 40 sessions offered at one campus and almost 30 at the other- are some students going to most of these)? If there are, does including these outliers influence the results? For example, we might think that a student who attends 10 workshops is more satisfied than a student who attends 2, but the student who attended 10 is represented 5x more in the data than the student who attended 2. Or, were the data collapsed to the individual level first (e.g. a student's responses to all questionnaires were averaged within that respondent) and then group means were calculated)? Having a better understanding of how many observations there are by respondent and the extent to which these results may be sensitive to outliers would be helpful.

Second, I think there are limitations of only have perceptions of satisfaction as an outcome and the descriptive design, but overall I think the authors have done a good job of motivating the study, explaining the theory of change behind these specific programs, and contextualizing their findings within the literature on mentoring. I would caution against making too broad of recommendations- for example, saying that because there aren't significant differences in satisfaction between age groups, we should get rid of age groupings in schools (lines 497-499). The lack of differences in this case could be because of selection or limited statistical power, so I think that conclusion might be beyond the scope of the findings presented. 

Author Response

RESPONSE TO REVIEWER 3 COMMENTS (ROUND 2)
I appreciate the authors' responsiveness to the reviews; I think this is an improved version
of the paper. However, I have two concerns that I think still need to be addressed.
1. First, and most importantly, I would still like a little more information about the number
of workshops students attended- there are 171 students and 157 questionnaires, which
is an average of about 3 sessions per student. The authors also note that 80% of students
attended more than 1 session. Is the distribution pretty tight around 3 sessions? Or are
there students who extend out to the tail (there were over 40 sessions offered at one
campus and almost 30 at the other- are some students going to most of these)? If there
are, does including these outliers influence the results? For example, we might think
that a student who attends 10 workshops is more satisfied than a student who attends
2, but the student who attended 10 is represented 5x more in the data than the student
who attended 2. Or, were the data collapsed to the individual level first (e.g. a student's
responses to all questionnaires were averaged within that respondent) and then group
means were calculated)? Having a better understanding of how many observations
there are by respondent and the extent to which these results may be sensitive to
outliers would be helpful.
Response 1: This study does not aim to analyze the general satisfaction of the mentee
with the program but with each of the workshops in which he has participated.
On the other hand, we would like to report that we have been sensitive to outliers.
We appreciate your comments and suggestions.
2. Second, I think there are limitations of only have perceptions of satisfaction as an
outcome and the descriptive design, but overall I think the authors have done a good job
of motivating the study, explaining the theory of change behind these specific programs,
and contextualizing their findings within the literature on mentoring. I would caution
against making too broad of recommendations- for example, saying that because there
aren't significant differences in satisfaction between age groups, we should get rid of
age groupings in schools (lines 497-499). The lack of differences in this case could be
because of selection or limited statistical power, so I think that conclusion might be
beyond the scope of the findings presented.
Response 2 : The phrase “This supports the need for schools to dismantle the rigid
criterion of age grouping and to stop seeing the role of the teacher as the main agent of
the learning process.” It has been replaced by “ These results encourage us to continue
working and addressing this issue in depth in order to investigate some of the data
obtained.”